# BRAIN2GAN; RECONSTRUCTING PERCEIVED FACES FROM THE PRIMATE BRAIN VIA STYLEGAN3

## ABSTRACT

Neural coding characterizes the relationship between stimuli and their correspond-
ing neural responses. The usage of synthesized yet photorealistic reality by gener-
ative adversarial networks (GANs) allows for superior control over these data: the
underlying feature representations that account for the semantics in synthesized
data are known a priori and their relationship is perfect rather than approximated
post-hoc by feature extraction models. We exploit this property in neural decoding
of multi-unit activity (MUA) responses that we recorded from the primate brain
upon presentation with synthesized face images in a passive fixation experiment.
First, the face reconstructions we acquired from brain activity were remarkably
similar to the originally perceived face stimuli. Second, our findings show that
responses from the inferior temporal (IT) cortex (i.e., the recording site furthest
downstream) contributed most to the decoding performance among the three brain
areas. Third, applying Euclidean vector arithmetic to neural data (in combination
with neural decoding) yielded similar results as on $w$-latents. Together, this pro-
vides strong evidence that the neural face manifold and the feature-disentangled
$w$-latent space conditioned on StyleGAN3 (rather than the $z$-latent space of ar-
bitrary GANs or other feature representations we encountered so far) share how
they represent the high-level semantics of the high-dimensional space of faces.

## 1 INTRODUCTION

The field of neural coding aims at deciphering the neural code to characterize how the brain rec-
ognizes the statistical invariances of structured yet complex naturalistic environments. *Neural en-
coding* seeks to find how properties of external phenomena are stored in the brain by modeling the
stimulus-response transformation (van Gerven, 2017). Vice versa, *neural decoding* aims to find
what information about the original stimulus is present in and can be retrieved from the measured
brain activity by modeling the response-stimulus transformation (Haynes & Rees, 2006; Kamitani
& Tong, 2005). In particular, reconstruction is concerned with re-creating the literal stimulus image
from brain activity. In both cases, it is common to factorize the direct transformation into two by

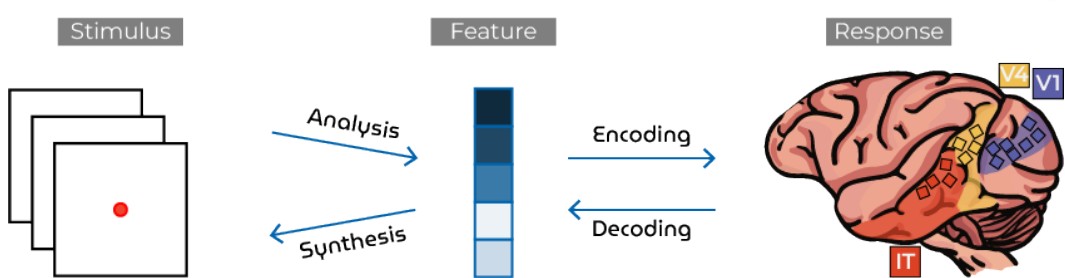

Figure 1: **Neural coding.** The transformation between sensory stimuli and brain responses via an
intermediate feature space. Neural encoding is factorized into a nonlinear "analysis" and a linear
"encoding" mapping. Neural decoding is factorized into a linear "decoding" and a nonlinear "syn-
thesis" mapping.

invoking an in-between feature space (Figure 1). Not only does this favor data efficiency as neural data is scarce but it also allows one to test alternative hypotheses about the relevant stimulus features that are stored in and can be retrieved from the brain.

The brain can effectively represent an infinite amount of visual phenomena to interpret and act upon the environment. Although such neural representations are constructed from experience, novel yet plausible situations that respect the statistics of the natural environment can also be mentally simulated or *imagined* (Dijkstra et al., 2019). From a machine learning perspective, generative models achieve the same objective: they capture the probability density that underlies a (very large) set of observations and can be used to synthesize new instances which appear to be from the original data distribution yet are suitably different from the observed instances. Particularly, generative adversarial networks (GANs) (Goodfellow et al., 2014) are among the most impressive generative models to date which can synthesize novel yet realistic-looking images (e.g., natural images and images of human faces, bedrooms, cars and cats (Brock et al., 2018; Karras et al., 2017; 2019; 2021) from randomly-sampled latent vectors. A GAN consists of two neural networks: a generator network that synthesizes images from randomly-sampled latent vectors and a discriminator network that distinguishes synthesized from real images. During training, these networks are pitted against each other until the generated data are indistinguishable from the real data. The bijective latent-to-image relationship of the generator can be exploited in neural decoding to disambiguate the synthesized images as visual content is specified by their underlying latent code (Kriegeskorte, 2015) and perform *analysis by synthesis* (Yuille & Kersten, 2006).

Deep convnets have been used to explain neural responses during visual perception, imagery and dreaming (Horikawa & Kamitani, 2017b;a; St-Yves & Naselaris, 2018; Shen et al., 2019b;a; Güçlütürk et al., 2017; VanRullen & Reddy, 2019; Dado et al., 2022). To our knowledge, the latter three are the most similar studies that also attempted to decode perceived faces from brain activity. (Güçlütürk et al., 2017) used the feature representations from VGG16 pretrained on face recognition (i.e., trained in a supervised setting). Although more biologically plausible, unsupervised learning paradigms seemed to appear less successful in modeling neural representations in the primate brain than their supervised counterparts (Khaligh-Razavi & Kriegeskorte, 2014) with the exception of (VanRullen & Reddy, 2019) and (Dado et al., 2022) who used adversarially learned latent representations of a variational autoencoder-GAN (VAE-GAN) and a GAN, respectively. Importantly, (Dado et al., 2022) used synthesized stimuli to have direct access to the ground-truth latents instead of using post-hoc approximate inference, as VAE-GANs do by design.

The current work improves the experimental paradigm of (Dado et al., 2022) and provides several novel contributions: face stimuli were synthesized by a feature-disentangled GAN and presented to a macaque with cortical implants in a passive fixation task. A decoder model was fit on the recorded brain activity and the ground-truth latents. Reconstructions were created by feeding the predicted latents from brain activity from a held-out test set to the GAN. Previous neural decoding studies

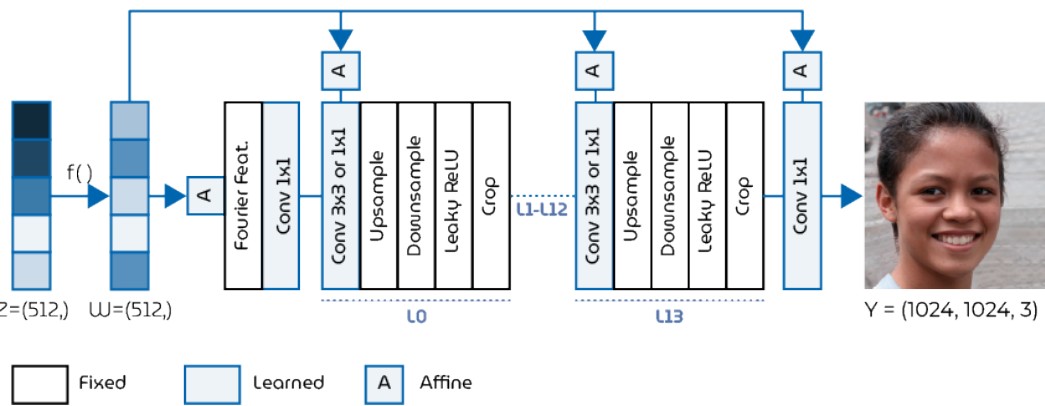

Figure 2: **StyleGAN3 generator architecture.** The generator takes a 512-dim. latent vector as input and transforms it into a $1024^2$ resolution RGB image. We collected a dataset of 4000 training set images and 100 test set images.

used noninvasive fMRI signals that have a low signal-to-noise ratio and poor temporal resolution leading to a reconstruction bottleneck and precluding detailed spatio-temporal analysis. This work is the first to decode photorealistic faces from intracranial recordings which resulted in state-of-the-art reconstructions as well as new opportunities to study the brain. First, the high performance of decoding via w-latent space indicates the importance of disentanglement to explain neural representations upon perception, offering a new way forward for the previously limited yet biologically more plausible unsupervised models of brain function. Second, we show how decoding performance evolves over time and observe that the largest contribution is explained by the inferior temporal (IT) cortex which is located at the end of the visual ventral pathway. Third, the application of Euclidean vector arithmetic to w-latents and brain activity yielded similar results which further suggests functional overlap between these representational spaces. Taken together, the high quality of the neural recordings and feature representations resulted in novel and unprecedented experimental findings that not only demonstrate how advances in machine learning extend to neuroscience but also will serve as an important benchmark for future research.

## 2 METHODS

### 2.1 DATA

#### 2.1.1 STIMULI

We synthesized photorealistic face images of $1024^2$ resolution from (512-dim.) $z$-latent vectors with the generator network of StyleGAN3 (Karras et al., 2020) (Figure 2) which is pretrained on the high-quality Flickr-Faces-HQ (FFHQ) dataset (Karras et al., 2019). The $z$-latents were randomly sampled from the standard Gaussian. First, StyleGAN3 maps the $z$-latent space via an 8-layer MLP to an intermediate (512-dim.) $w$-latent space in favor of feature disentanglement. That is, the original $z$-latent space is restricted to follow the data distribution that it is trained on (e.g., older but not younger people wear eyeglasses in the training set images) and such biases are entangled in the $z$-latents. The less entangled $w$-latent space overcomes this such that unfamiliar latent elements can be mapped to their respective visual features (Karras et al., 2019). Second, we specified a truncation of 0.7 so that the sampled values are ensured to fall within this range to benefit image quality. During synthesis, learned affine transformations integrate $w$-latents into the generator network with adaptive instance normalization (like *style transfer* (Huang & Belongie, 2017)) as illustrated in Figure 2. Finally, we synthesized a training set of 4000 face images that were each presented once to cover a large stimulus space to fit a general model. The test set consisted of 100 synthesized faces that were averaged over twenty repetitions.

#### 2.1.2 FEATURES

For the main analysis, the $z$- and $w$-latent space of StyleGAN3 were both used as the in-between feature space. In addition, we also extracted the intermediate layer activations to our face stimuli from alexnet for object recognition (Krizhevsky, 2014), VGG16 for face recognition (Parkhi et al., 2015) and the discriminator network of StyleGAN3. We fit multiple encoding models to see how well their feature representations can explain the recorded responses. Because the features from VGGFace and the discriminator were very large ($\sim 10^6$), we performed downsampling, as done in (Eickenberg et al., 2017). I.e., for each channel in the activation, the feature map was spatially smoothed with a Gaussian filter and subsampled such that the total number of output features was lower than 50,000 per image. The kernel size was set to be equal to the downsampling factor.

#### 2.1.3 RESPONSES

We recorded multi-unit activity (Super & Roelfsema, 2005) with 15 chronically implanted electrode arrays (64 channels each) in one macaque (male, 7 years old) upon presentation with the synthesized face images in a passive fixation experiment (Figure 3). Neural responses were recorded in V1 (7 arrays), V4 (4 arrays) and IT (4 arrays) leading to a total of 960 channels (see electrode placings in Figure 1). For each trial, we averaged the early response of each channel using the following time-windows: 25-125 ms for V1, 50-150 ms for V4 and 75-175 ms for IT. The data was normalized as in (Bashivan et al., 2019) such that for each channel, the mean was subtracted from all the responses which were then divided by the standard deviation. All procedures complied with the NIH Guide

Figure 3: **Passive fixation task.** The monkey was fixating a red dot with gray background for 300 ms followed by a fast sequence four face images ($500^2$ pixels): 200 ms stimulus presentation and 200 ms inter-trial interval. The stimuli were slightly shifted to the lower right such that the fovea corresponded with pixel (150, 150). The monkey was rewarded with juice if fixation was kept for the whole sequence.

for Care and Use of Laboratory Animals and were approved by the local institutional animal care and use committee.

## 2.2 MODELS

We used linear mapping to evaluate our claim that the feature- and neural representation effectively encode the same stimulus properties, as is standard in the field. A more complex nonlinear transformation would not be valid to support this claim since this could theoretically map anything to anything. We used regularization for encoding due to the high dimensionality of the feature space.

### 2.2.1 DECODING

Multiple linear regression was used to model how the individual units within feature representations $y$ (e.g., $w$-latents) are linearly dependent on brain activity $\mathbf{x}$ per electrode:

$$\mathcal{L} = \frac{1}{2} \sum_{i=1}^{N} \left( y_i - \mathbf{w}^T \mathbf{x}_i \right)^2 \tag{1}$$

where $i$ ranges over samples. This was implemented by prepending a dense layer to the generator architecture to transform brain responses into feature representations which were then run through the generator as usual. This response-feature layer was fit with ordinary least squares while the remainder of the network was kept fixed. Note that no truncation was applied for the reconstruction from predicted features/latents.

### 2.2.2 ENCODING

Kernel ridge regression was used to model how every recording site in the visual cortex is linearly dependent on the stimulus features. That is, an encoding model is defined for each electrode. In contrast to decoding, encoding required regularization to avoid overfitting since we predicted from feature space $\mathbf{x}_i \rightarrow \phi(\mathbf{x}_i)$ where $\phi()$ is the feature extraction model. Hence we used ridge regression where the norm of $\mathbf{w}$ is penalized to define encoding models by a weighted sum of $\phi(\mathbf{x})$:

$$\mathcal{L} = \frac{1}{2} \sum_{i=1}^{N} \left( y_i - \mathbf{w}^T \phi(\mathbf{x}_i) \right)^2 + \frac{1}{2} \lambda_j ||\mathbf{w}||^2 \tag{2}$$

where $\mathbf{x} = (\mathbf{x}_1, \mathbf{x}_2, ..., \mathbf{x}_N)^T \in \mathbb{R}^{N \times d}$, $y = (y_1, y_2, ..., y_N)^T \in \mathbb{R}^{N \times 1}$, $N$ the number of stimulus-response pairs, $d$ the number of pixels, and $\lambda_j \geq 0$ the regularization parameter. We then solved for $\mathbf{w}$ by applying the "kernel trick" (Welling, 2013):

$$\mathbf{w_j} = (\lambda_j \mathbf{I}_m + \Phi\Phi^T)^{-1} \Phi y \tag{3}$$

where $\Phi = (\phi(x_1), \phi(x_2), ..., \phi(x_N)) \in \mathbb{R}^{N \times q}$ (i.e., the design matrix) where $q$ is the number of feature elements and $y = (y_1, y_2, \ldots, y_N \in \mathbb{R}^{N \times 1}$. This means that $\mathbf{w}$ must lie in the space induced by the training data even when $q \gg N$. The optimal $\lambda_j$ is determined with grid search, as in (Güçlü & van Gerven, 2014). The grid is obtained by dividing the domain of $\lambda$ in $M$ values and evaluate

model performance at every value. This hyperparameter domain is controlled by the capacity of the model, i.e., the effective degrees of freedom dof of the ridge regression fit from $[1, N]$:

$$\text{dof}(\lambda_j) = \sum_{i=1}^{N} \frac{s_i^2}{s_i^2 + \lambda_j} \qquad (4)$$

where $s$ are the non-zero singular values of the design matrix $\Phi$ as obtained by singular value decomposition. We can solve for each $\lambda_j$ with Newton's method. Now that the grid of lambda values is defined, we can search for the optimal $\lambda_j$ that minimizes the 10-fold cross validation error.

### 2.3 EVALUATION

Decoding performance was evaluated by six metrics that compared the stimuli from the held-out test set with their reconstructions from brain activity: latent similarity, alexnet perceptual similarity (object recognition), VGG16 perceptual similarity (face recognition), latent correlation, pixel correlation and structural similarity index measure (SSIM). For *latent similarity*, we considered the cosine similarity per latent dimension between predicted and ground-truth latent vectors:

$$\text{Latent cos similarity} = \frac{\hat{z}_i \cdot z_i}{\sqrt{\sum_{i=1}^{512}(\hat{z}_i)^2}\sqrt{\sum_{i=1}^{512}(z_i)^2}}$$

where $\hat{z}$ and $z$ are the 512-dimensional predicted and ground-truth latent vectors, respectively. For *perceptual similarity*, we computed the cosine similarity between deeper layer activations (rather than pixel space which is the model input) extracted by deep neural networks. Specifically, we fed the stimuli and their reconstructions to alexnet pretrained on object recognition (Krizhevsky, 2014) and VGG16 pretrained on face recognition (Parkhi et al., 2015) and extracted the activations of their last convolutional layer. We then considered the cosine similarity per activation unit:

$$\text{Perceptual cos similarity} = \frac{f(\hat{x})_i \cdot f(x)_i)}{\sqrt{\sum_{i=1}^{n}(f(x)_i)^2}\sqrt{\sum_{i=1}^{n}(f(\hat{x})_i)^2}}$$

where $x$ and $\hat{x}$ are the $224 \times 224$ RGB (image dimensionality that the models expects) visual stimuli and their reconstructions, respectively, $n$ the number of activation elements, and $f(.)$ the image-activation transformation. *Latent- and pixel correlation* measure the standard linear (Pearson product-moment) correlation coefficient between the latent dimensions of the predicted and ground-truth latent vectors and the luminance pixel values of stimuli and their reconstructions, respectively. *SSIM* looked at similarities in terms of luminance, contrast and structure (Wang et al., 2004).

Furthermore, we introduce a new *SeFa attribute similarity* metric between stimuli and their reconstructions using the intrinsic semantic vectors of the generator which we accessed using closed-form factorization ("SeFa") (Shen & Zhou, 2021). In short, the unsupervised SeFa algorithm decomposes the pre-trained weights of the generator into 512 different latent semantics (of 512 dimensions each) which can be used for editing the synthesized images in $w$-space. This is also a means to understand what each latent semantic encodes: if a face becomes younger or older when traversing the latent in the negative or positive direction of the latent semantic, we can conclude post-hoc that it encodes the attribute "age". In our case, we used it to score each stimulus and reconstruction by taking the inner product between their $w$-latent and latent semantic and check for similarity.

### 2.4 IMPLEMENTATION DETAILS

We used the original PyTorch implementation of StyleGAN3 (Karras et al., 2021), the PyTorch implementation of alexnet and the keras implementation of VGG16. All analyses were carried out in Python 3.8 on the internal cluster.

## 3 RESULTS

### 3.1 NEURAL DECODING

We performed neural decoding from the primate brain via the feature-disentangled $w$-latent space of StyleGAN3; see Figure 4 and Table 1 for qualitative and quantitative results, respectively. Perceptually, it is obvious that the stimuli and their reconstructions share a significant degree of similarity

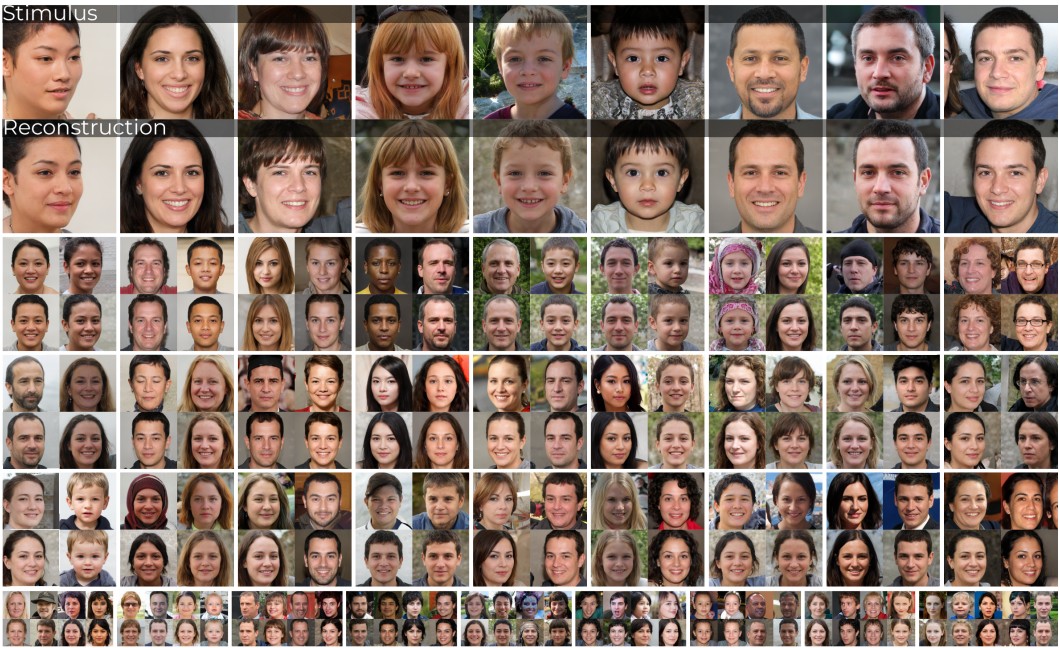

Figure 4: **Qualitative results:** The 100 test set stimuli (top row) and their reconstructions from brain activity (bottom row).

(e.g., gender, age, pose, haircut, lighting, hair color, skin tone, smile and eyeglasses). The importance of feature disentanglement for neural decoding is highlighted when compared to the decoding performance via the original $z$-latent space. The qualitative results from $z$-latent space can be found in Appendix A.1. We provided a visual guide for the six evaluation metrics in Appendix A.4 by showing the stimulus-reconstruction pairs with the five highest- and the five lowest similarity for each metric. In addition, we also repeated the experiment with another macaque that had silicon-based electrodes in V1, V2, V3 and V4; see Appendix A.3.

The contribution of each brain region to the overall reconstruction performance was determined by occluding the recordings from the other two sites. Concretely, the two site responses were replaced with the average response of all but the corresponding response such that only the site of interest remained. Alternatively, one could evaluate region contribution by training three different decoders

Table 1: **Quantitative results.** The upper block decoded the $z$-latent whereas the lower block decoded the $w$-latent from brain activity. Decoding performance is quantified in terms of six metrics: latent cosine similarity, latent correlation (Student's t-test), perceptual cosine similarity using alexnet and VGGFace, pixel-wise correlation (Student's t-test) and structural similarity index (SSIM) in pixel space between stimuli and their reconstructions ($mean \pm std.error$). To make the comparison more fair, the predicted $z$-latents were transformed to $w$-space and truncated at 0.7 for comparison with the ground-truth $w$-latents. The rows display performance when using either the recordings from "all" recording sites or only from a specific brain area. The latter is achieved by occluding the recordings from the other two brain regions in the test set. Neural decoding from all brain regions into the $w$-latent resulted in the overall highest reconstruction performance.

| | | Lat. sim. | Lat. corr. | Alexnet sim. | VGG16 sim. | Pixel corr. | SSIM |
|---|---|---|---|---|---|---|---|
| $z$ | All | $0.3909 \pm 0.0070$ | $0.2824 \pm 0.0054$ | $0.2416 \pm 0.0023$ | $0.1789 \pm 0.0013$ | $0.4331 \pm 0.0001$ | 0.3811 |
| | V1 | $0.2785 \pm 0.0079$ | $0.1398 \pm 0.0051$ | $0.1871 \pm 0.0022$ | $0.1430 \pm 0.0012$ | $0.2887 \pm 0.0001$ | 0.2640 |
| | V4 | $0.2763 \pm 0.0079$ | $0.1362 \pm 0.0050$ | $0.1956 \pm 0.0022$ | $0.1485 \pm 0.0012$ | $0.2430 \pm 0.0001$ | 0.2211 |
| | IT | $0.3012 \pm 0.0076$ | $0.1747 \pm 0.0055$ | $0.2054 \pm 0.0022$ | $0.1498 \pm 0.0012$ | $0.3105 \pm 0.0001$ | 0.2794 |
| $w$ | All | $\mathbf{0.4579 \pm 0.0076}$ | $\mathbf{0.2908 \pm 0.0047}$ | $\mathbf{0.2740 \pm 0.0029}$ | $\mathbf{0.2391 \pm 0.0017}$ | $\mathbf{0.6055 \pm 0.0001}$ | **0.5547** |
| | V1 | $0.3792 \pm 0.0089$ | $0.1478 \pm 0.0047$ | $0.1447 \pm 0.0023$ | $0.1151 \pm 0.0012$ | $0.2958 \pm 0.0001$ | 0.2256 |
| | V4 | $0.3783 \pm 0.0023$ | $0.1450 \pm 0.0049$ | $0.1856 \pm 0.0020$ | $0.1315 \pm 0.0011$ | $0.1816 \pm 0.0001$ | 0.1684 |
| | IT | $0.4009 \pm 0.0085$ | $0.1828 \pm 0.0051$ | $0.1861 \pm 0.0026$ | $0.1451 \pm 0.0014$ | $0.4119 \pm 0.0001$ | 0.3275 |

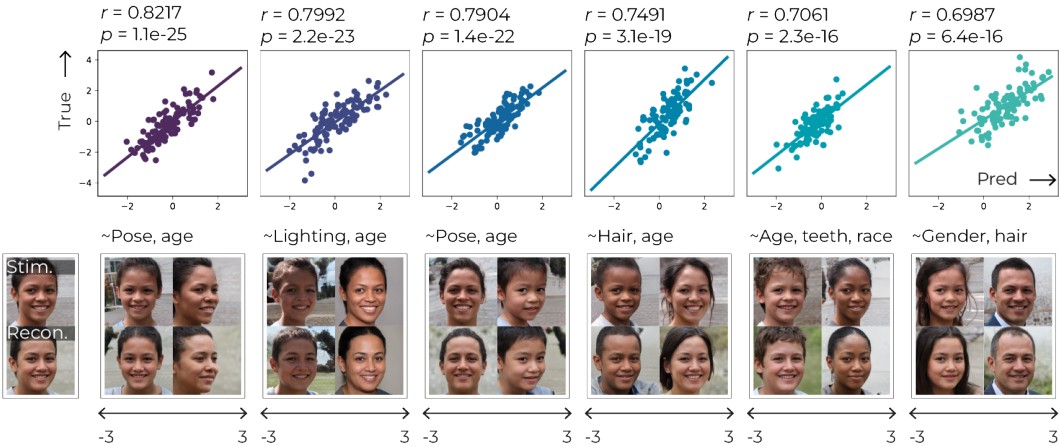

Figure 5: **SeFa attribute similarity.** The SeFa attribute scores were computed for stimuli and their reconstructions and evaluated for similarity in terms of correlation (Student's t-test). The six plots display the attribute scores of the 100 stimuli (True; Y-axis) and the predictions (Pred; X-axis) for the six latent semantics with the highest similarity. We travel in the semantic direction and edit an arbitrary face to reveal what facial attributes a latent semantic encodes. That is, a latent semantic was subtracted of or added to a $w$-latent which was fed to the generator to create the corresponding face image. For instance, when the semantic boundary with the highest similarity ($r = 0.8217$) is subtracted from a $w$-latent, the corresponding face changes pose to the left and becomes younger whereas its pose changes to the right and gets older when this semantic is added.

on neural data subsets, but the current occlusion method made it more interpretable how a region contributed to the same decoder's performance. As such, we found that this is for the largest part determined by responses from IT - which is the most downstream site we recorded from.

We validated our results with a permutation test as follows: for a thousand times, we sampled a hundred random latents from the same distribution as our original test set and generated their corresponding face images. Per iteration, we checked whether these random latents and faces were closer to the ground-truth latent and faces than our predicted latents and faces. We found that our predicted latents from brain activity and corresponding faces were always closer to the original stimuli for the $w$-latents and all six metrics, yielding statistical significance ($p < 0.001$). This indicates that our high decoding performance is not just a consequence of the high-quality images that the generator synthesizes. The charts showing the six similarity metrics over iterations for the random samples and our predictions based on brain activity can be found in Appendix A.2.

Next, we quantified how well facial attributes were predicted from brain activity (Figure 5). The 512 latent semantics are known to be hierarchically organized (Shen & Zhou, 2021) and we also find this back in our predictive performance where the highest and lowest correlations are found at the earlier

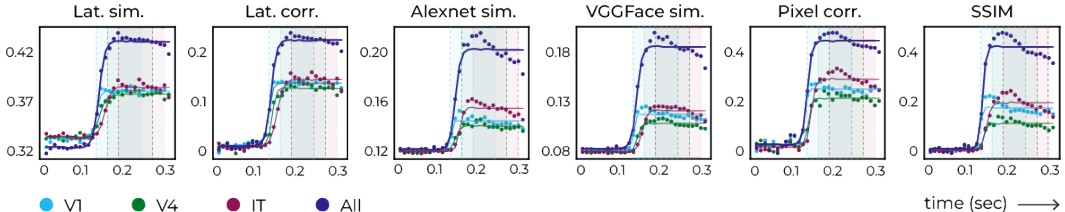

Figure 6: **Neural decoding performance in time**. The development of decoding performance (Y-axis) in terms of all six metrics based on the recorded response data in V1, V4, IT and all brain regions and over the full time course of 300 ms (X-axis). Stimulus onset happened at 100 ms. For visualization purposes, a sigmoid was fit to these data points and the shaded areas denote the predefined time windows we used for our original analysis.

and later latent semantics, respectively. Face editing revealed that the earlier latent semantics encode clear and well-known facial attributes (e.g., gender, age, skin color, lighting and pose) whereas those in later latent semantics remain unclear since editing did not result in perceptual changes.

Finally, Figure 6 shows how decoding performance evolved in time. Rather than taking the average response over the predefined time windows for V1, V4 and IT, we took the average response over a 10 ms window that was slid without overlap over the full time course of 300 ms. This resulted in thirty responses over time per stimulus. As expected, performance climbed and peaked first for (early) V1, then (intermediate) V4 and lastly (deep) IT. We can see that IT outperformed the other two regions in terms of all six metrics.

## 3.2 NEURAL ENCODING

Neural encoding predicted the brain activity from the eight (feature) layers from alexnet for object recognition, VGG16 for face recognition and the StyleGAN3 discriminator (i.e., $3 \times 8 = 24$ encoding models in total). Encoding performance was assessed by correlating the predicted and recorded responses (Student's t-test) after which the model layer with the highest performance was assigned to each recording site on the brain (Figure 7A). Our results show a gradient from early to deeper brain areas for all three models. That is, visual experience is partially determined by the selective responses of neuronal populations along the visual ventral "what" pathway (Ungerleider & Mishkin, 1982) such that the receptive fields of neurons in early cortical regions are selective for simple features (e.g., local edge orientations (Hubel & Wiesel, 1962)) whereas those of neurons in deeper regions respond to more complex patterns of features (Gross et al., 1972; Hung et al., 2005). Previous work has shown how the features extracted by deep convolutional neural networks predict neural responses in the ventral visual stream to perceived naturalistic stimuli in the human brain (Yamins et al., 2014; Cadieu et al., 2014; Khaligh-Razavi & Kriegeskorte, 2014; Güçlü & van Gerven, 2015; Yamins & DiCarlo, 2016; Cichy et al., 2016; Eickenberg et al., 2017) as well as in the primate brain (Freiwald & Tsao, 2010; Chang & Tsao, 2017). In line with the literature, our results show that (early) V1 encodes earlier model layers whereas (deeper) IT encodes deeper model layers.

## 3.3 WALKING THE NEURAL FACE MANIFOLD VIA W-LATENT SPACE

Manifold hypothesis states that real-world data instances can be viewed as points in high-dimensional space that are concentrated on manifolds which (locally) resemble Euclidean space. Linear changes in the (low-dimensional) GAN latent landscape directly translate to the corresponding (high-dimensional) pixel space and thus approximate local manifolds (Shao et al., 2018). That is, visual data that look perceptually similar in terms of certain features are also closely positioned in latent space. As such, interpolation between two distinct latent vectors resulted in an ordered set

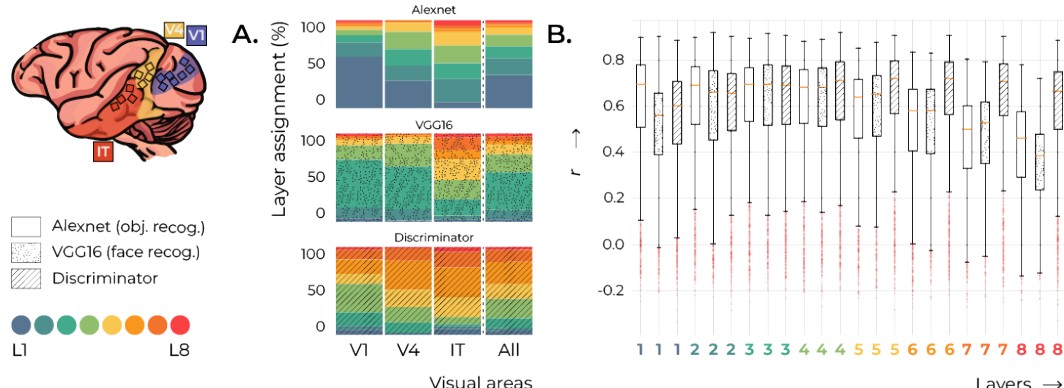

Figure 7: **A.** Layer assignment across all electrodes to visual areas V1, V4 and IT. We observe a gradient for Alexnet, VGG16 and the discriminator where early layers are mostly assigned to V1, intermediate layers to V4 and deep layers to IT. **B.** Performance of all 24 encoding models (x-axis) in terms of correlation (Student's t-test, y-axis) between all predicted and recorded brain responses shows that encoding performance was good overall.

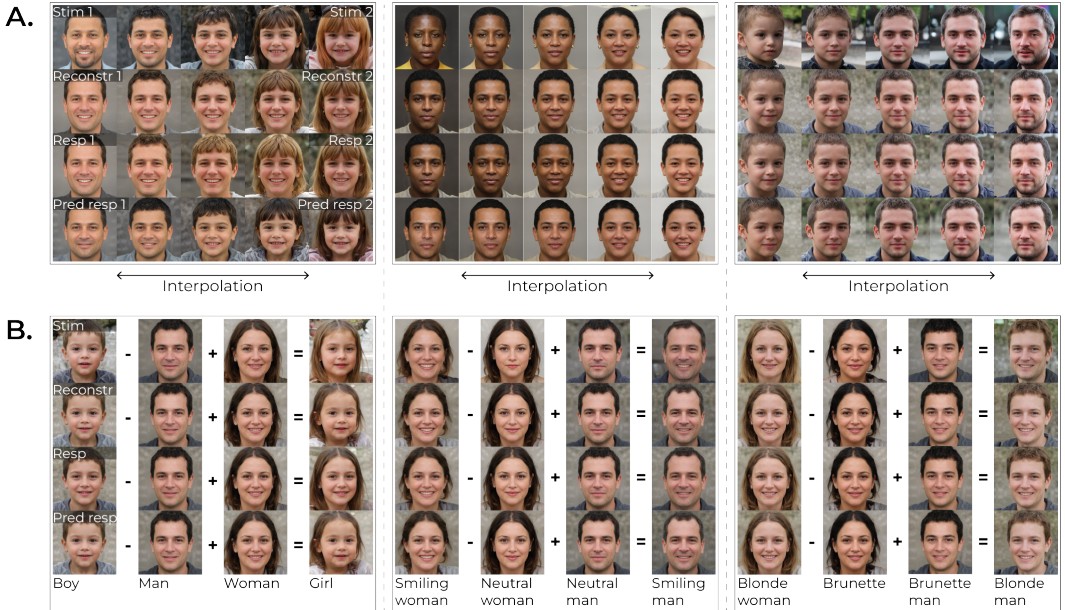

Figure 8: Linear operations were applied to ground-truth $w$-latents (row 1), $w$-latents decoded from brain activity (row 2), recorded responses which we then decoded into $w$-latents (row 3) and encoded responses from ground-truth $w$-latents which we then decoded $w$-latents. **A.** Linear interpolation between the $w$-latents of two test set stimuli. **B.** Vector arithmetic with the average $w$-latent of specific facial attributes. To obtain the average $w$-latent we averaged as many relevant samples as possible from the test set.

of latents where contained semantics in the corresponding images vary gradually with latent code. We used spherical rather than regular linear interpolation to account for the spherical geometry of the latent space (i.e., multimodal Gaussian distribution). The latent space also obeys simple arithmetic operations (Mikolov et al., 2013). The generated faces from interpolation (Figure 8A) and vector arithmetic (Figure 8B) in neural space were perceptually similar as when applied directly to the $w$-latent space. This indicates that the functional neural face manifold and $w$-latent space are organized similarly such that responses can be linearly modified to obtain responses to unseen faces.

# 4 DISCUSSION

Neural decoding of brain activity during visual perception via the feature-disentangled $w$-latent space conditioned on StyleGAN3 resulted in image reconstructions that strongly resemble the originally-perceived stimuli, making it the state-of-the-art in the field. Although it is safe to assume that the brain represents the visual stimuli it is presented with, it has been largely unsolved *how* it represents them as there are virtually infinite candidate representations possible to encode the same image. The goal of this study was to find the correct representation. Our results demonstrate that StyleGAN3 features/latents are linearly related to brain responses such that latent and response must encode the same real-world phenomena similarly. This indicates that StyleGAN3 successfully disentangled the neural face manifold (DiCarlo & Cox, 2007) rather than the conventional $z$-latent space of arbitrary GANs or any other feature representation we encountered so far. Note that StyleGAN3 has never been optimized on neural data. We also found that the features of the discriminator are predictive of neural responses. The StyleGAN3-brain correspondence can shed light on what drives the organization of (neural) information processing in vision. For instance, the analogy between adversarial training of StyleGAN3 and predictive coding where the brain is continuously generating and updating its mental model of the world to minimize prediction errors. To conclude, unsupervised generative modeling can be used to study biological vision which in turn supports the development of better computational models thereof and other (clinical) applications.

## 5 ETHICS STATEMENT

In conjunction with the evolving field of neural decoding grows the concern regarding mental privacy. Because we think it is likely that access to subjective experience will be possible in the foreseeable future, we want to emphasize that it is important to at all times strictly follow the ethical rules and regulations regarding data extraction, storage and protection. It should never be possible to invade subjective contents of the mind.

## 6 REPRODUCIBILITY STATEMENT

Upon publication, we will make our data publicly available together with the code and a detailed description on how to recreate our results to ensure transparency and reproducibility.

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
