# OpenReview forum: "Brain2GAN; Reconstructing perceived faces from the primate brain via StyleGAN3"
_ICLR.cc/2023/Conference — Submitted to ICLR 2023_

### Official Review · Reviewer_1pMd · 2022-10-24

**Confidence:** 4
**Correctness:** 3
**Technical Novelty And Significance:** 2
**Empirical Novelty And Significance:** 2
**Recommendation:** 3

**Clarity, Quality, Novelty And Reproducibility:**

The experiments are not based on a public dataset and no implementation was released, which makes difficulty of reproduction.

**Strength And Weaknesses:**

Strength:The author finds evidence that the neural face manifold and the disentangled w-latent space conditioned on StyleGAN3 are similar in how they represent the high-level semantics of the high-dimensional space of faces. Especially, the StyleGAN3 has never been optimized on neural data.

Weaknesses:
1) Why the author use a AlexNet pretrained on object recognition task rather than on the face recognition task? What is the different between the cosine similarity from the AlexNet and the VGGFace?
2) For the conclusions in the summary, “This provides strong evidence that the neural face manifold and the disentangled w-latent space conditioned on StyleGAN3 (rather than the z-latent space of arbitrary GANs or other feature representations we encountered so far) share how they represent the high-level semantics of the high-dimensional space of faces”, the author should add some qualitative comparison to make a visual support. Also, the qualitative comparison can give the reader a visual feeling of the difference in image space caused by the difference in six decoding performance evaluation metrics.
3) The gap between the z-latent and w-latent in the “Alexnet sim” and “VGGFace sim” are much more lower than in the “Lat. sim.” and the “Lat. corr.”, how does the author explain this phenomenon? How much does the difference in “Alexnet sim” and “VGGFace sim” affect the reconstruction image quality? If the visualized difference is not obvious, it will weaken the conclustion that “This provides strong evidence that the neural face manifold and the disentangled w-latent space conditioned on StyleGAN3 (rather than the z-latent space of arbitrary GANs or other feature representations we encountered so far) share how they represent the high-level semantics of the high-dimensional space of faces” in the abstract.


**Summary Of The Paper:**

The author found that the neural responses of macaques to facial visual stimuli are very similar to the w-space of styleGAN3. Specifically, the neural response of the macaques' brain can be decoded into w-space using linear mapping, and the corresponding generation faces are similar to the perceived stimuli. Also, the reconstructing looks similar when applying the same linear operations on the brain activity and the latent.

**Summary Of The Review:**

This paper provides evidence that the neural face manifold and the disentangled w-latent space conditioned on StyleGAN3 are similar in how they represent the high-level semantics of the high-dimensional space of faces. But more comparison are suggest to be added between the reconstruction quality of z-latent and w-latent.

---

> ### Author Response · Authors · 2022-11-15
> **We would like to thank the reviewer for the helpful comments. We sincerely believe that they have helped us improve our manuscript. We have carefully addressed them as detailed below in our point-by-point responses.**
>
> **Why the author use a AlexNet pretrained on object recognition task rather than on the face recognition task? What is the different between the cosine similarity from the AlexNet and the VGGFace?**
>
> We use the features from alexnet pretrained on object recognition as well as the features from VGG16 pretrained on face recognition to evaluate similarity of stimuli and their reconstructions. Earlier studies have shown that object recognition models are the most accurate to explain neural representations during visual perception, so we include this metric in our analysis for completeness. We can explain the difference between alexnet and VGG16 similarity because our study uses face images and alexnet models more generic features whereas VGG16 is better at detecting differences in facial features.
>
> **For the conclusions in the summary, the author should add some qualitative comparison to make a visual support. Also, the qualitative comparison can give the reader a visual feeling of the difference in image space caused by the difference in six decoding performance evaluation metrics.**
>
> We agree with the reviewer that the qualitative reconstructions from z-space together with a visual guide regarding perceptual similarity would be a valuable addition to the manuscript. The reconstructions from z-latent space can be found in Appendix A.1 and the visual guide in Appendix A.4.
>
> **The gap between the z-latent and w-latent in the “Alexnet sim” and “VGGFace sim” are much more lower than in the “Lat. sim.” and the “Lat. corr.”, how does the author explain this phenomenon? How much does the difference in “Alexnet sim” and “VGGFace sim” affect the reconstruction image quality? If the visualized difference is not obvious, it will weaken the conclustion that “This provides strong evidence that the neural face manifold and the disentangled w-latent space conditioned on StyleGAN3 (rather than the z-latent space of arbitrary GANs or other feature representations we encountered so far) share how they represent the high-level semantics of the high-dimensional space of faces” in the abstract.**
> **More comparison are suggest to be added between the reconstruction quality of z-latent and w-latent.**
>
> The different metrics have different scales and dimensionality, and are thus not directly comparable among each other. That being said, we have now included how different metrics rank the reconstructions (Appendix A.4) as well as the reconstructions from z-latent space (Appendix A.1), showing obvious differences in reconstruction accuracy.
>
> **The experiments are not based on a public dataset and no implementation was released, which makes difficulty of reproduction.**
>
> The complete dataset will be released upon publication so that the experiment can be reproduced and used as a benchmark for future studies.

---

### Official Review · Reviewer_4n7A · 2022-10-25

**Confidence:** 2
**Correctness:** 3
**Technical Novelty And Significance:** 2
**Empirical Novelty And Significance:** 2
**Recommendation:** 5

**Clarity, Quality, Novelty And Reproducibility:**

The experiments and writing are clear and sound. It might be hard to reproduce the result, since it needs a macaque.


**Strength And Weaknesses:**

Pro:
The idea to evaluate StyleGAN3 W space and Z space by using a macaque is interesting.
The experiments are extensive and solid.
Con:
I believe this paper is more experimental. There is not much novelty.


**Summary Of The Paper:**

This paper is about "Neural coding", that is to characterize the biological brain recognization. They used a StyleGAN3 to synthesis several high solution images, and show those images to a 7 years old macaque , and recorded its multi-unit brain activity. Finally, they use several latent similarity metric to compute distant z and w space. Finally, they conclude that StyleGAN3 W space is better than Z space in decoding performance.

**Summary Of The Review:**

It is my first time to  know "neural coding" field.
The experiments that to explore brain actively with neural representation is interesting.
Although the there is a face that StyleGAN3 W space is more disentangled than Z space, the experimental shows its closer relationship to brain perceived stimuli is exciting.

---

> ### Author Response · Authors · 2022-11-15
> **We would like to thank the reviewer for the helpful comments. We sincerely believe that they have helped us improve our manuscript. We have carefully addressed them as detailed below in our point-by-point responses.**
>
> **I believe this paper is more experimental. There is not much novelty.**
>
> We clarified the methodological and empirical novelty of our contributions by including the following section in the introduction of the updated manuscript:
>
> *“Deep convnets have been used to explain neural responses during visual perception, imagery and dreaming (Horikawa & Kamitani, 2017b;a; St-Yves & Naselaris, 2018; Shen et al., 2019b;a; Gucluturk et al., 2017; VanRullen & Reddy, 2019; Dado et al., 2022). To our knowledge, the latter three are the most similar studies that also attempted to decode perceived faces from brain activity. Gucluturk et al. (2017) used the feature representations from VGG16 pretrained on face recognition (i.e., trained in a supervised setting). Although more biologically plausible, unsupervised learning paradigms seemed to appear less successful in modeling neural representations in the primate brain than their supervised counterparts (Khaligh, 2014) with the exception of VanRullen & Reddy (2019) and Dado et al. (2022) who used adversarially learned latent representations of a variational autoencoder-GAN (VAE-GAN) and a GAN, respectively. Importantly, Dado et al. (2022) used synthesized stimuli to have direct access to the ground-truth latents instead of using post-hoc approximate inference, as VAE-GANs do by design.*
>
> *The current work improves the experimental paradigm of Dado et al. (2022) and provides several novel contributions: face stimuli were synthesized by a feature-disentangled GAN and presented to a macaque with cortical implants in a passive fixation task. A decoder model was fit on the recorded brain activity and the ground-truth latents. Reconstructions were created by feeding the predicted latents from brain activity from a held-out test set to the GAN. Previous neural decoding studies used noninvasive fMRI signals that have a low signal-to-noise ratio and poor temporal resolution leading to a reconstruction bottleneck and precluding detailed spatio-temporal analysis. This work is the first to decode photorealistic faces from intracranial recordings which resulted in state-of-the-art reconstructions as well as new opportunities to study the brain. First, the high performance of decoding via w-latent space indicates the importance of disentanglement to explain neural representations upon perception, offering a new way forward for the previously limited yet biologically more plausible unsupervised models of brain function. Second, we show how decoding performance evolves over time and observe that the largest contribution is explained by the inferior temporal (IT) cortex which is located at the end of the visual ventral pathway. Third, the application of Euclidean vector arithmetic to w-latents and brain activity yielded similar results which further suggests functional overlap between these representational spaces. Taken together, the high quality of the neural recordings and feature representations resulted in novel and unprecedented experimental findings that not only demonstrate how advances in machine learning extend to neuroscience but also will serve as an important benchmark for future research.”*
>
> **It might be hard to reproduce the result, since it needs a macaque.**
>
> The data and code will be shared upon publication which will be used as a benchmark for future research and will be one of the largest and highest-quality publicly available datasets of its kind.
>
> **Some of the paper’s claims have minor issues. A few statements are not well-supported, or require small changes to be made correct.**
>
> We double-checked our claims and updated them where necessary. We are happy to apply more specific changes to any remaining issues when pointed out.

---

### Official Review · Reviewer_1cBk · 2022-10-29

**Confidence:** 3
**Correctness:** 3
**Technical Novelty And Significance:** 2
**Empirical Novelty And Significance:** 1
**Recommendation:** 6

**Clarity, Quality, Novelty And Reproducibility:**

- originality: many previous works should be mentioned (see above section). As it stands, I don't think this paper adds a lot of novelty.
- clarity: Clarity is good. Some figures (figure 7) have hard to read portions.
- reproducability: The authors have said they will release the data, including the macaque data and code
- quality: The reconstructed images are impressive. But confidence in the significance of these results could be improved by a more complete discussion of the permutation test results. (See below)

# questions/minor comments
- figure 8B: typo - "Brunet" --> "Brunette man"
- Section 2.3.2 $Y_i$ is defined, but I can't see where it is ever used?
- figure 7A: Hard to see the texture of VGG16
- If it's simple to produce, could we see a plot of all electrode locations?
- The authors mention a permutation test in section 3.1? Could we see the full results of the permutation test? What is the average closeness of the random latent vector to the ground truth?

**Strength And Weaknesses:**

# strengths
- New macaque monkey data.
- This seems like the first attempt at reconstructing faces from intracranial data. (Is it?) If so, that should be mentioned somewhere
- The reconstructed images are impressive

# weaknesses
- I have concerns about the originality of this work. Decoding high quality faces from brain signals has been previously accomplished (Dado et al. 2021, VanRullen and Reddy 2019, Güçlütürk and Güçlü 2017). The decoding and encoding techniques, namely the use of GAN latent spaces, have already been discussed in previous works. In this work, the authors use a slightly different model (StyleGAN3 vs StyleGAN), but the overall methods remain the same.
- The analysis of reconstructed faces in terms of attributes is also introduced in these previous works. (The authors do acknowledge this)
- I think the paper could be much improved if the relationship with previous works were more clearly explained.
- The novel contribution of this work seems mainly to be the newly collected data. But even this is only done for one subject.
- The difference between the $w$ latent space and $z$ latent space could be explained more clearly. It's mentioned that the $w$ latent space is obtained by passing the $z$ latent space through an MLP, but since it results in much better reproductions (relative to $z$), could the authors explain a little bit more about what makes the two spaces different?

## references
- Dado et al. 2021 Hyperrealistic neural decoding: Reconstructing faces from fMRI activations via the GAN latent space (https://www.biorxiv.org/content/10.1101/2020.07.01.168849v3.full)
- VanRullen and Reddy 2019 Reconstructing faces from fMRI patterns using deep generative neural networks https://www.nature.com/articles/s42003-019-0438-y
- Güçlütürk and Güçlü et al. 2017 Reconstructing perceived faces from brain activations with deep adversarial neural decoding

**Summary Of The Paper:**

This paper studies the encoding and decoding of faces in the (macaque) brain. To study decoding, they generate faces by sampling a random vector from the latent space of a GAN. They show the faces to a macaque monkey and record the neural response. A linear model is trained to map the neural response to the latent vector that produced the face image. On test data, they find that the recovered vector can be used to produce a high quality image of a face that matches the original face qualitatively and quantitatively. To study encoding, they map the activations of a trained vision model (AlexNet, VGG16) to the neural response. They find that correlation with the neural response increases with depth in the machine model.

**Summary Of The Review:**

The reconstruction results are impressive, but have already been accomplished in many previous works. The decoding techniques are not significantly different than what was introduced in prior work, although the data is new. But even then, the data consists only of neural recordings for a single subject.

---

> ### Author Response · Authors · 2022-11-15
> **We would like to thank the reviewer for the helpful comments. We sincerely believe that they have helped us improve our manuscript. We have carefully addressed them as detailed below in our point-by-point responses.**
>
> **First attempt at reconstructing faces from intracranial data? Concerns about originality. Relationship with previous works must be more clearly explained. Not a lot of novelty.**
>
> We now added the following section under the introduction:
>
> _“Deep convnets have been used to explain neural responses during visual perception, imagery and dreaming (Horikawa & Kamitani, 2017b;a; St-Yves & Naselaris, 2018; Shen et al., 2019b;a; Gucluturk et al., 2017; VanRullen & Reddy, 2019; Dado et al., 2022). To our knowledge, the latter three are the most similar studies that also attempted to decode perceived faces from brain activity. Gucluturk et al. (2017) used the feature representations from VGG16 pretrained on face recognition (i.e., trained in a supervised setting). Although more biologically plausible, unsupervised learning paradigms seemed to appear less successful in modeling neural representations in the primate brain than their supervised counterparts (Khaligh, 2014) with the exception of VanRullen & Reddy (2019) and Dado et al. (2022) who used adversarially learned latent representations of a variational autoencoder-GAN (VAE-GAN) and a GAN, respectively. Importantly, Dado et al. (2022) used synthesized stimuli to have direct access to the ground-truth latents instead of using post-hoc approximate inference, as VAE-GANs do by design._
>
> _The current work improves the experimental paradigm of Dado et al. (2022) and provides several novel contributions: face stimuli were synthesized by a feature-disentangled GAN and presented to a macaque with cortical implants in a passive fixation task. A decoder model was fit on the recorded brain activity and the ground-truth latents. Reconstructions were created by feeding the predicted latents from brain activity from a held-out test set to the GAN. Previous neural decoding studies used noninvasive fMRI signals that have a low signal-to-noise ratio and poor temporal resolution leading to a reconstruction bottleneck and precluding detailed spatio-temporal analysis. This work is the first to decode photorealistic faces from intracranial recordings which resulted in state-of-the-art reconstructions as well as new opportunities to study the brain. First, the high performance of decoding via w-latent space indicates the importance of disentanglement to explain neural representations upon perception, offering a new way forward for the previously limited yet biologically more plausible unsupervised models of brain function. Second, we show how decoding performance evolves over time and observe that the largest contribution is explained by the inferior temporal (IT) cortex which is located at the end of the visual ventral pathway. Third, the application of Euclidean vector arithmetic to w-latents and brain activity yielded similar results which further suggests functional overlap between these representational spaces. Taken together, the high quality of the neural recordings and feature representations resulted in novel and unprecedented experimental findings that not only demonstrate how advances in machine learning extend to neuroscience but also will serve as an important benchmark for future research.”_
>
> **The difference between the w-latent space and z-latent space could be explained more clearly.**
>
> We now added the following explanation under 2.2.1 Stimuli:
>
> _“That is, the original z-latent space is restricted to follow the data distribution that it is trained on (e.g., older but not younger people wear eyeglasses in the training set images) and such biases are entangled in the z-latents. The less entangled w-latent space overcomes this such that unfamiliar latent elements can be mapped to their respective visual features [Karras, 2019]."_
>
> **The novel contribution of this work seems mainly to be the newly collected data. But even this is only done for one subject.**
>
> We now performed the experiment with another macaque as well (Appendix A.3). As such, we report the quantitative results for two subjects which is common in the field.
>
> **Some figures (figure 7) have hard to read portions.**
>
> We now changed the texture of VGG16 and showed the outcomes in three distinct graphs.
>
> **Could we see the full results of the permutation test?**
>
> Yes, we are currently re-running the permutation analyses to report the average closeness of random vectors to the ground-truth vectors and will update the manuscript with these results.
>
> **figure 8B: typo - "Brunet" --> "Brunette man"**
>
> This suggestion is now incorporated.
>
> **Section 2.3.2 Yi is defined, but I can't see where it is ever used?**
>
> We now double-checked all the defined variables and corrected them where necessary.
>
> **Plot of all electrode locations?**
>
> We now added a schematic illustration showing the electrode placings (Fig. 3).

---

> > ### Author Response · Authors · 2022-11-15
> > **Response to point about attribute similarity**
> >
> > **The analysis of reconstructed faces in terms of attributes is also introduced in these previous works.**
> >
> > The reviewer is right that an attribute similarity metric has been used by [Dado et al., 2022]. Importantly, this metric was based on the decision boundaries identified by SVMs in a supervised setting. The attribute similarity in our work is based on the intrinsic latent semantics of the generator weights that are extracted by the unsupervised SeFa algorithm [Shen & Zhou, 2021], which makes it more straightforward to use due to the lack of label requirements. As such, we introduce a new "SeFa" attribute similarity metric for neural decoding.

---

> > > ### Author Response · Authors · 2022-11-17
> > > **Response to point about permutation test**
> > >
> > > **Could we see the full results of the permutation test?**
> > >
> > > We now included plots with the six similarity metrics over iterations for randomly sampled latents/faces as well as our predictions from brain activity (Appendix A.2). The random samples are never closer to the ground-truth than our predictions which indicates that our high decoding performance is not just a consequence of the high-quality images by StyleGAN.

---

> ### Comment · Reviewer_1cBk · 2022-11-28
> **Reply to authors**
>
> I thank the authors for their response.The revised version of the paper has additional subject data and clarification about the relationship with previous work. I think this is an improvement. The authors also clarify that they are the first to apply their method to intracranial data. I will recommend acceptance and update my score to a 6. My main concern is with the novelty for the ML community. From what I can tell, the main takeaway is that neural decoding can be done better with better off-the-shelf representations, i.e., from StyleGAN3. The downstream consequences of this finding for ML researchers seems limited.

---

### Decision · Program_Chairs · 2023-01-20

**Decision:**

Reject

**Justification For Why Not Higher Score:**

The method is of limited interest to the ICLR community. A venue where the key contribution is not the idea, but any improvements in performance, would be a much better fit.


**Justification For Why Not Lower Score:**

N/A

**Metareview: Summary, Strengths And Weaknesses:**

Summary: Decoding neural activity while subjects are presented with images into the w-latent space of StyleGAN3.

Strengths: A new dataset. The first reconstruction of faces from intracranial data.

Weaknesses: Limited novelty and a lack of clarity about what the ICLR community can learn from this work. It's unclear how the dataset could be reused, it's fairly small (2 subjects), and designed very specifically for this experiment. An audience that would appreciate the value of decoding faces from intracranial data on its own would be more appropriate and receptive.